# Affinity, Specificity, and Cooperativity of DNA Binding by Bacterial Gene Regulatory Proteins

**DOI:** 10.3390/ijms23010562

**Published:** 2022-01-05

**Authors:** Jannette Carey

**Affiliations:** 1Chemistry Department, Princeton University, Princeton, NJ 08544, USA; jcarey@princeton.edu; 2Laboratory of Structural Biology and Bioinformatics, Institute of Microbiology, Czech Academy of Sciences, 37333 Nové Hrady, South Bohemia, Czech Republic

**Keywords:** biological constraints, host–guest chemistry, pre-organization, cryptic thermodynamic factors, protein folding coupled to ligand binding, gestalt properties of proteins, drug design

## Abstract

Nearly all of biology depends on interactions between molecules: proteins with small molecules, proteins with other proteins, nucleic acids with small molecules, and nucleic acids with proteins that regulate gene expression, our concern in this Special Issue. All those kinds of interactions, and others, constitute the vast majority of biology at the molecular level. An understanding of those interactions requires that we quantify them to learn how they interact: How strongly? With which partners? How—and how well—are different partners distinguished? This review addresses the evolution of our current understanding of the molecular origins of affinity and specificity in regulatory protein–DNA interactions, and suggests that both these properties can be modulated by cooperativity.

## 1. Molecular Interactions Run Biology

To understand biology on a molecular level, we need to understand the biology of its molecules. Nearly all of biology depends on interactions between molecules: proteins with small molecules, proteins with other proteins, nucleic acids with small molecules, and nucleic acids with proteins that regulate gene expression, our concern in this Special Issue. All those kinds of interactions, and others, constitute the vast majority of biology at the molecular level. An understanding of those interactions requires that we quantify them to learn how they interact: How strongly? With which partners? How—and how well—are different partners distinguished?

These questions are embodied in the terms *affinity* and *specificity*. Every reaction has characteristic values of affinity and specificity. We quantify affinity—the strength of binding—using ΔG, the free energy change between the bound and free (unbound) states of the partners. ΔG tells us how much the free energy of the system decreases upon binding (for cases where binding is spontaneous and ΔG is favorable for complex formation, thus negative) compared to the unbound state of the system. We derive ΔG from measurement of the equilibrium constant for the binding reaction, K_eq_, through the relationship ΔG = −RT ln K_eq_. For reversible binding reactions like those considered here, in which no covalent alteration occurs, K_eq_ can be determined equivalently in the association (K_a_) or dissociation (K_d_ = 1/K_a_) direction.

Where affinity appears familiar and straightforward, specificity appears to be more elusive, subject to interpretation in different ways in different contexts or by different authors, as indeed it is in the literature. Some definitions are inconsistent with each other and/or not thermodynamically sound. However, there is one way to quantify specificity that is straightforward, thermodynamically sound, and entirely general, and should be in much wider use: the *difference* in free energy change, ΔΔG, between ΔG for binding of ligand A with target B and ΔG for binding of ligand A’ with target B. This is a thermodynamically rigorous definition of specificity, just like the ΔG values on which it is based [1,2]. However, it is an operational definition in the sense that it depends on the identities of ligands A and A’, and its value therefore cannot be considered to be an inherent property of target B. (To be philosophical about it, we would be strictly correct in saying that a fundamental property of *specificity* is that no target has a fixed value of specificity.)

The reader will have noticed by now the unsubtle segue from biology—in vivo—to biochemistry—implicitly and explicitly in vitro. Although values of affinity and specificity measured in vitro are valid for the conditions in which they are determined, those conditions typically differ dramatically from conditions in vivo, which in fact can be known at best only approximately. Thus, there is in general no practical way to relate in vitro values to in vivo values. We assume, reasonably, but mostly without evidence, that at least in a relative way the values determined in vitro reflect values that would pertain in vivo. Furthermore, the discussion here considers only *equilibrium* properties of the interactions, not anything about their kinetics, although many biological systems operate far from equilibrium. Nevertheless, there *is* insight that can be derived from an analysis under these limitations, as well as some exceptions and qualifications to them, as will be described. Collectively, the insights that can be derived about biology from analyses under in vitro conditions reflect roots in physical chemistry that in turn may derive from the “unreasonable effectiveness of mathematics in the natural sciences” [3].

## 2. Molecular Origins of Affinity and Specificity

One point of view that has proved useful for thinking about the values of affinity and specificity is that they reflect biological constraints that are imposed on each molecular interaction in order for it to serve its intended purpose in the organism. This viewpoint led my research group some years ago to introduce a two-dimensional (2D) plane or map displaying the combined values of affinity and specificity for various biological interactions [4]. An updated view of this map is shown on Figure 1A, where affinity and specificity are represented as axes from lower to higher relative values, with affinity in the x dimension (x chosen arbitrarily) and specificity in the y dimension. On this plane in Figure 1 are plotted a number of different molecular interactions in biology (placed to reflect approximate relative values of affinity and specificity), in colored boxes to indicate pairs that are related to each other, and one ringer that is not derived from biology, as described below.

Our aim with the 2D map was to provide a framework for the research question of interest to us, What molecular mechanisms are used to achieve given combinations of affinity and specificity? This question was beautifully addressed in early work on the so-called host–guest systems of synthetic organic chemistry, the one map entry on Figure 1A that is non-biological. The implications of that work are profoundly informative for any interaction including biological ones, as recognized by the 1987 Nobel prize in chemistry to Jean-Marie Lehn, Donald J. Cram, and Charles J. Pedersen (see, e.g., the Nobel lecture publication by Cram (1988) [5]). Thus, we now take a steep but essential dive into chemical history that will inform us as we return—asymptotically—to the biological question.

### 2.1. Forgotten Lessons from the Host–Guest Chemistry 

The Nobel Prize was awarded not for the novel synthetic compounds *per se*, but for the profound insights into molecular recognition that were derived from the work. It was discovered early on that certain kinds of oligomers of a general form like that shown in Figure 2A could bind ionic species (here the generic monovalent cation M^+^) despite presenting no obvious sites for coordination of an ion. In the case shown, each monomeric unit joined by single bond to its neighbors is composed of a benzene ring with a methyl ether group on one end and a methyl group on the other end. Each curved arrow indicates relevant rotational degrees of freedom about the single bonds between each pair of benzene rings and between each ether group and its benzene ring.

What the authors found was that upon binding to a monovalent cation, flexible compounds like the one shown form a cavity enclosing the ion, as shown on the right in Figure 2A. An original aim was to design compounds like the one shown here, called hosts, that would have the ability to bind compounds like the ion, called guests. The general idea was to design hosts that could bind guests selectively and with high affinity. Among the expected practical applications was, for example, use in an industrial process that might require two different cations in successive steps, but the first ion inhibits the second process. After the first step, a host could be added to bind up the first ion, enabling addition of a second ion to permit the process to proceed with the second step. This design was aimed at the specificity of binding. Another aim was to have these binding events proceed with very high affinity so that the host could be used in very small quantity that would therefore be economical, and thus not interfere in subsequent steps itself. This design was aimed at the affinity of binding.

These early hosts, like the one shown in Figure 2A, which were named ‘podands,’ had extremely low affinity, as shown by the K_a_ values indicated. The podands also displayed no detectable specificity, having indistinguishable affinities for Li^+^, Na^+^, and K^+^, ions of very different ionic radii, charge density, and hydration that ought to make them distinguishable if the host were doing what it was designed to do. The truly brilliant insight of the early workers in this field rested on their recognition that in the bound state the host forms a cavity enclosing the bound ion. The reasoning that they applied at this point is really why they won the Nobel Prize; it has laid the groundwork for much of organic chemistry ever since, and the underlying concept has spread to all branches of chemistry. They reasoned that if the bound state looks like it surrounds the ion, then a host that looks more like the bound state *before* binding the guest ought to bind better. That is exactly what they found when they circularized the host in its unbound state to form what they termed a ‘spherand.’

### 2.2. The Profound Effects of Pre-Organization

Figure 2B shows the host–guest complex formed by the circularized host. The 2D host structure looks unchanged at first glance, but close inspection shows that on the right panels a new single bond at the upper left of the host joins the two terminal benzene rings, enabling the host to have the intended circular shape already in its free state. (Although the shape is apparently circular in a 2D representation, in three dimensions this spherand has a cup-like shape, as represented by the cartoon in Figure 1A, where two spherands enclose a triangular cartoon guest.) Thus, the second-generation hosts were ‘pre-organized’ in the bound conformation, a term that is now common throughout chemistry and biochemistry, although the knowledge has largely been lost that both the term and its underlying principle originate in the host–guest chemistry. The later hosts that were designed to be pre-organized in the bound conformation displayed affinity *and* specificity that were profoundly altered, just as intended. Lithium was now bound with an association equilibrium constant K_a_ greater than 10^16^ M^−1^. That is a gain of *12 orders of magnitude* in affinity. Additionally, now sodium was bound with an affinity constant of 10^14^ M^−1^. Potassium was still bound weakly, around 10^4^ M^−1^. These values indicate that the circularized host can discriminate between lithium and potassium by *12 orders of magnitude*, and between sodium and potassium by 10 orders of magnitude. Thus, specificity has increased enormously, in addition to the huge gains in affinity.

How is this dramatic result possible simply by adding *just one single bond* between the two benzene rings to close the linear free host into the circular conformation? You may well wonder if the structure of the complex is somehow altered by this small change. The answer to this is unquestionably NO. The *only* difference in *bonding* in the two complexes is the single new bond closing the host into a circular form; there are *no* differences in *intermolecular* bonding between the host and the guest. *The bonds between the host and the ion guests are identical for the linear and circular hosts.* In other words, from looking at the structures, even at atomic resolution with x-ray diffraction of crystals, you would not be able to distinguish the interactions between the host and guest in the two cases, nor correlate one complex with far higher affinity and specificity than the other. This is really a stunning result that is completely counterintuitive. Instinctively we imagine that an interaction with much higher affinity and specificity must have different bonding between its partners than one with weak affinity and nonexistent specificity. Yet, that is not true here: the bonding between the host and guest is *identical* for both the linear and circular hosts.

This surprising finding indicates that structural analysis alone cannot help us understand the profound differences in affinity and specificity between the linear and circular hosts. We must turn to dynamics and thermodynamics to discover the underlying reasons for these enormous differences. From the thermodynamic viewpoint, however, drastically altered binding despite identical intermolecular bonding was neither unexpected nor inexplicable, and here is why. The rotational motions enjoyed by the linear host in the unbound state must be severely—essentially entirely—damped in order to form a circular arrangement around the bound guest. The damping out of all those motions upon binding represents a very large entropic cost that accompanies the binding process. This unfavorable entropic cost opposes binding, and thus diminishes the extent of free energy lowering upon binding. A thermodynamically accurate, if colloquial, way to think of this situation is that with the linear host the entropic cost is ‘paid’ during the binding process, and the payment comes out of the binding free energy. In contrast, in the circularized host, those rotational motions are already excluded in the unbound state, eliminating that source of entropic cost during the binding process, and eliminating its unfavorable contribution to the binding free energy as well.


*(As a thought experiment, consider that in the case of the circularized host the rotational motions are in fact damped, just as they were during binding in the case of the linear host, but in the circular case they are damped **prior** **to** the binding event. Therefore, the entropic cost of damping the motions must still be somehow paid; where **exactly** is this cost paid in the case of the circularized host? At what step(s), and in what form(s)?)*


### 2.3. Rational Design of Affinity and Specificity

Let us look at some of the other principles that were used in this example to achieve the intended rational design of affinity and specificity. Recall that the goal was to maximize *both* affinity *and* specificity so as to make these hosts useful for industrial processes. One obvious strategy was to build in functional groups in order to provide bonding between the host and the intended guest. That was of course already implemented in the linear host, where in the bound state the oxygen atoms of the methyl ether groups are all pointing inward, providing a localized region of electronegativity that complements the cationic guest. Note that for the *unbound* linear host to adopt the *bound* conformation (i.e., in the absence of the ion) will require that it overcome not only the many degrees of freedom in the unbound host, but also the very substantial unfavorable contribution to free energy due to the expected electrostatic repulsion from all those methyl ethers pointing toward each other but uncompensated by a bound ion. Thus, the bound conformation is expected to be a very minor component of the population of conformational states of the free host at equilibrium.

Another factor that played an important role in manipulating affinity is not at all obvious from looking at the structures of the free or bound states. The researchers built in sites where components of the solvent could be bound to the host in its free state, and be released upon binding of the guest. Why is this counterintuitive strategy useful? Solvent molecules that are pre-bound to the free host and displaced upon binding of the guest experience a big gain in entropy, reflecting the difference in degrees of freedom between being relatively more fixed to the host vs. being free in the bulk solvent. This increase in entropy contributes favorably to the overall entropy change of the whole system upon binding, and thus to the free energy lowering. The researchers showed that solvent components can be manipulated to make favorable contributions to guest binding. This example shows how our thinking about interaction processes is often biased toward looking only at the molecules that are of direct interest, those whose structures are directly visualized. It is easy to forget that thermodynamically it is the entire system that is relevant to the changes in free energy that accompany binding, and not just the molecules we care about or can visualize directly.

Finally, the most obvious thing they did was to add that one single bond to the host in the free state, which had two important consequences. The first was to greatly reduce entropy losses upon ion binding by rigidifying the host in its free state, as we saw above. However, rigidification by circularization restricts mobility not only of the entire host but also of the host’s functional groups, which point precisely into the central site as required for interaction with a bound ion. Closing the ring of the free host fixes the size of the central cavity, which may or may not match the size of a guest ion. The rigidity acts to limit access and/or distort bonding by large ions, and to make bonding distances unfavorably long for smaller ions. In contrast, in the linear host, the functional groups have some ability to adapt and accommodate different guests, a fact that was manifested in the finding that the linear hosts were unable to discriminate between lithium and sodium, unlike the closed-ring hosts that have size-match selectivity, reflected in their very high specificity values.

By using these methods, the researchers found that affinity and specificity accrue approximately in parallel. In other words, the molecular mechanisms they employed to increase affinity also increase specificity. This is reflected in the position of the host–guest systems on the 2D map at the extreme top right of the diagonal of the plot in Figure 1A. However, biology populates the entire affinity/specificity map, and puts demands on interactions that require map positions very different from the upper right corner. This means that biology requires molecular mechanisms that permit *independent* modulation of affinity and specificity, unlike those used in host–guest chemistry.

### 2.4. Bonding Cannot Be Equated with Binding

Let us summarize by reiterating what is probably the single most important point from these lessons of the host–guest chemistry. *The bonds between the partners are **identical** whether the host is pre-organized or not, yet both the affinity and specificity are orders of magnitude different.* This fact leads to the following profound conclusion that applies generally to all molecular interactions: bonding between the partners does not predict their affinity, and in fact is scarcely even related to affinity; this can be shorthanded as *bonding does not predict binding*. An implication of this fact is that structural analysis of bonding between partners cannot inform us about their affinity. The basic reason is that structural analysis of intermolecular bonds neglects all the rest of the system and its many contributions, both favorable and unfavorable, to both enthalpy and entropy of the system. This author has called those features *cryptic* contributions to the binding process, because they are hidden from our view except through the lens of thermodynamics. Computational approaches are getting better at accounting for some of these features, but it is still likely that successful predictions of affinity are relatively rare. It is impossible to know how rare they are, however, because we have no public repository for failed predictions. Thus, the successes are celebrated, but may be less common than they seem. Furthermore, we do not always know all the sources of contributions from the system that need to be accounted for in calculations. One of the few ways we have of identifying cryptic contributions is to study interactions quantitatively under different solution conditions. In one of the best-studied examples, the affinity of many protein–DNA interactions was found to depend on the presence of salt in the buffer, leading to important insights that are discussed further below.

The host–guest example reminds us that a complete understanding of any molecular system requires integration of structural and thermodynamic viewpoints. Partly because thermodynamics is so abstract, and structures are so seductive because we relate so well to their visual content, it is instinctive to focus on structures and bonding. Furthermore, thermodynamics stresses system properties even though in some cases, perhaps many in biology, we cannot describe the system fully or accurately. So, logically, we focus on the things that we can see and neglect those we cannot see or describe. However, thermodynamically it is the whole system that matters, and the things that we see may not even be those that dominate the overall energetic picture. Without integrating structure and thermodynamics, we cannot understand why some complexes form in preference to others—the essence of biology.

Only rarely do students of ligand binding today know about host–guest chemistry. Its important lessons seem to have been largely lost from mainstream general chemistry since the time of the Nobel Prize, which is a pity because those lessons are so profound. It may seem like old history by now, but it is a very important part of history for understanding molecular recognition, a field that has grown enormously in scope and significance since the time of Lehn, Cram, and Pedersen, reflecting their fundamental contributions.

### 2.5. Summary: Cryptic Contributions to Binding

The principal source of the enormous gain in affinity by the spherand is that pre-organization reduces the huge entropy loss that accompanies binding by the non-pre-organized podand host. The gain in specificity arises from the fixed constraint on the size of the binding cavity in the pre-organized host. These surprising results should serve to remind us that structural analysis alone cannot explain the molecular origins of affinity and specificity for any system. We must marry structures with dynamics and thermodynamics to discover the underlying reasons for given values.

Among the relevant insights for biology was the enormous extent to which thermodynamic driving forces favoring the bound state can be derived from cryptic sources, those that are not detected directly in structural analysis of the bound or free states. The most compelling practical consequence of this principle from the host–guest work was the conceptualization and implementation of pre-organization, which is now used routinely throughout synthetic organic chemistry, and is relevant as well in biology, as we shall see.

## 3. Case Study: The Human Immune System

Returning to biology, the lavender-shaded interactions on Figure 1A derive from the human immune system, with antibody–antigen interactions in the top middle, and MHC–peptide interactions in the lower right. In antibody–antigen interactions affinity needs to be high because once an antibody has captured a provoking antigen, it has to deliver that antigen to the downstream steps of the immune system to initiate an immune response. It would be counterproductive if the complex did not survive to do those downstream steps, so affinity must be high to ensure the antigen is retained. On the other hand, specificity has to be very high to avoid a possible immune response to an autoantigen that may resemble the provoking antigen. So to do its job the immune system must produce antibodies capable of achieving high affinity and extraordinarily high specificity toward new antigens that may resemble autoantigens. The human immune system has indeed evolved this capability through use of strategies at the molecular, cellular, and organismal levels including somatic recombination, somatic mutation, affinity maturation, clonal selection, and immune cell recruitment (for review, see [6]).

In contrast to the antibody–antigen case, in the lower right-hand corner we have the example of the MHC–peptide complex, another molecular interaction from the same human immune system. MHC, the major histocompatibility complex, is a protein of the immune system whose job is to bind to every peptide that is processed by proteolysis out of every protein antigen encountered in a human lifetime. Unlike antibodies, which are elaborated freely as needed in response to every new antigen encountered, each person has a fixed and very limited repertoire of MHC molecules, about 10 or 12 of them, that nevertheless must be able to recognize every conceivable peptide that is processed out of every conceivable protein antigen that is ever encountered in a lifetime. This means that each MHC must have the ability to bind peptides promiscuously, i.e., with low specificity, consistent with its map position. However, MHCs still have to bind their target peptides with high affinity because, just like in the antibody case, once the MHC protein has captured a peptide, the complex must survive to initiate the next steps in the immune cascade.

These two requirements for the MHC–peptide complex appear to be in contradiction in demanding the counterintuitive combination of high affinity and low specificity. Yet, evolution has produced MHCs with exactly the ability to bind peptides strongly but with very little preference among different peptide sequences. The MHC–peptide case is one of few in which the molecular mechanisms behind this map position are understood in some detail [7,8]. From a bird’s eye view, the answer is that the free MHC protein exists in an incompletely structured state—debate continues whether it qualifies as a molten globule, but naming it is not essential to understand the mechanism. Only upon binding a peptide does the MHC protein conform to the expectations of unique structure and defined stability typically associated with a fully-folded protein. Furthermore, consistent with its high affinity, dissociation of the peptide occurs only rarely—only when the MHC–peptide complex unfolds—because the peptide has become essentially part of the overall protein structure, which unfolds infrequently, like other ‘stable’ proteins.

How do these features comport with low specificity, with very few peptide positions restricted even as to amino acid residue type, let alone sequence? The answer to this mystery can be deduced from folding studies of other mutant proteins. In the early days of directed mutagenesis it was considered surprising that so many mutations could be accommodated while maintaining a protein’s structure and function. The surprise probably was rooted in conceptions of proteins as little rocks, a seemingly logical but misleading inference from protein crystallography that neglected their existence as dynamic objects. Although the dynamic nature of proteins was recognized early on by the most prescient experimentalists [9,10], crystallographic studies of proteins dominated the view for decades. Eventually, it became widely understood that protein folds are inherently adaptable to sequence variety, as in fact they must be to sustain evolution itself. In fact, this conclusion could have been reached far earlier from the mid-twentieth-century observation that dysfunctional mutant proteins of bacteria or phages are easily rescued by pseudoreversion [11,12,13,14]. Second-site revertants show that function, and thus by inference also structure, can be maintained in proteins by long-range mutual adjustments. Thus, MHC–peptide binding need not be sequence-specific as long as the bound peptide can contribute to a stable fold of the complex.

These immune system examples illustrate that biology demands very different map locations—combinations of affinity and specificity—for various molecular interactions, even for those that operate in the same biological system. In the case of the immune system, even though the molecules come from the same biological system their biological demands are different, and therefore their placement on the map is different. The review of principles presented here serves again to reemphasize the need to combine structural and thermodynamic analyses of molecular interactions.

## 4. Protein–DNA Interactions

In red shading on Figure 1A, two kinds of DNA-binding proteins are shown that further illustrate the point that biology demands many different, sometimes seemingly impossible or contradictory, combinations of affinity and specificity, even for molecules involved in closely related biological processes. Regulon regulators, in the middle of the map, are DNA-binding proteins in bacteria that are responsible for coordinating gene expression from related but non-contiguous genes or operons. As will be discussed in detail, their typically modest values of both affinity and specificity represent a special case among regulatory proteins. In contrast, the nonspecific DNA-binding proteins present in all organisms, in the lower left-hand corner, bind DNA promiscuously to package it by compaction into chromatin or nucleoids. This group also includes proteins that facilitate the action of certain DNA metabolic enzymes such as DNA or RNA polymerases, which require the DNA duplex to be disrupted into single strands. To do their jobs for biology, nonspecific DNA-binding proteins must be able to recognize all DNA irrespective of its sequence. Therefore, they bind with low specificity. However, they also must bind with low affinity because they must release the DNA when it is needed for another purpose like replication or transcription.

It is well known from structural studies that the molecular interactions at the protein–DNA interfaces of these two kinds of proteins, regulon regulators and nonspecific DNA-binding proteins, are not recognizably different. Yet, they occupy different map locations because the biological demands on the molecules are distinct, just as in the antibody–antigen and MHC–peptide case. These comparisons lead us to conclude that nature is selecting for *optimal* combinations of affinity and specificity, and not for *maximal* values of either parameter. If maximal values were being selected by nature, then all interactions would be in the upper right with host–guest systems. The fact that the 2D map is broadly populated implies that natural selection operates on affinity and specificity, as seems obvious. The two parameters are not necessarily under selection separately; selection pressure seems likely to be on the combined values. Although no direct evidence can be adduced for or against that view, the logic argument can be made that gains in affinity have the potential to alter specificity, so the two parameters may need to be controlled in a coordinated, or at least mutually responsive, manner.

### 4.1. Thermodynamics of Molecular Interactions

The host–guest example leads us to expect that the thermodynamics of molecular interactions can be both complex and surprising, and that bonding between the partners need not correlate with the affinity or specificity of binding. An early example from protein–DNA interactions will illustrate the point and bring additional perspectives. The binding of the bacteriophage lambda Cro protein to its several DNA operator sites was found to have very strong affinity, but ΔH was large and positive, as was ΔS [15]. Yet, crystal structures showed ample bonding of many kinds between the partners: ionic interactions, van der Waals contacts, and apparent hydrogen bonds (inferred from heavy-atom distances and angles). At first it was natural to doubt the verity of the thermodynamic results, but they have been upheld for Cro, and extended to other examples as well. The thermodynamic results—positive values of ΔH—say that the observed bonds between the partners are NOT what is holding the complex together! What can we say then about why the Cro–DNA complex forms at all? For this answer we must consider the thermodynamic results in their entirety, and focus on the fact that they reflect the entire system. When we do that, we realize that the reason the complex forms is because the bound state has lower total free energy (i.e., is more favorable) than the collective free states of the components. How can that be? Early solution experiments with Cro and other DNA-binding proteins provided some hints.

Many of these proteins have very strong dependence of their DNA affinity on the concentration of salt in solution, as alluded to above. Yet, counterions are not observed in structures of free or bound DNA in numbers that could account for the magnitude of this effect, and quantitative study of the dependence ruled out simple electrostatic effects to explain the dependence [16]. Theoretical analysis suggested that DNA, as a polyelectrolyte, attracts a large number of counterions that partially neutralize the negative charge, and it was proposed that these ions are associated with DNA in a diffuse way rather than a site-bound way [17,18]. These counterions were considered to be “thermodynamically bound”, enabling them to contribute to the overall energetics of binding while being unseen by structural methods—in other words, although they cannot be detected in structures, their presence is reflected in the salt concentration dependence—a classical cryptic contributing factor. These inferences led to the proposal that counterions displaced from the DNA surface upon binding of a protein make a large favorable entropic contribution to the overall free energy of binding when the ions change from a relatively more localized state on the free DNA to a completely delocalized state after being displaced into the bulk solution upon formation of the complex. This model, parameterized using experimental values for the salt dependence of *lac* repressor protein–DNA binding, was even able to explain the complex kinetics of the protein’s motions on DNAs of various lengths during the search for its specific operator target [19].

This example shows that completely cryptic processes that need not be reflected in the observed bonding between the partners can dominate the overall free energy of binding. Returning to the Cro example with its positive ΔH and negative ΔG, we can see why the bonds between the partners need not correlate with affinity. In fact, careful thought will indicate that even in cases where ΔH *is* negative, its value cannot be correlated with the bonds between the partners. In other words, one validated example of a thermodynamic profile like that of Cro, of which there are by now many other examples not confined to DNA-binding proteins, indicates that in *no* case can the bonds between partners be considered as the reason a complex is held together. This leads to the general conclusion that there is no simple reconciliation of structure with thermodynamics; their relationship is complex; cryptic, even devious, contributions may be missing from the structural picture but may dominate the thermodynamics; and surprises abound.

### 4.2. The Special Case of Regulatory Protein–DNA Interactions

DNA-binding proteins involved in regulating gene expression (center in Figure 1), on the other hand, must recognize DNA sequences (operators) at typically several different recognition sites of a regulon in order to bring all regulon members under common metabolic control. Studies of many of these proteins have shown that the operator DNA sequences bound by a given regulator are seldom identical, and in the most extreme cases are hardly recognizable as members of the regulon. This observation indicates that specificity must be very finely tuned to permit recognition of the full range of true regulon sequences while rejecting non-members whose sequences may be only slightly more different. On the other hand, the affinity of these proteins often has a middle value as well, seemingly lower than expected considering that they typically contact their DNA targets over a surface area much larger than those between proteins and small molecules, some of which have substantially higher affinities. Regulon regulators have to bind strongly enough to execute the gene control function they are designed for, but they cannot bind so strongly that they do not release the DNA when it is required for some other process like transcription. So they need to have a middle value of affinity and a middle value of specificity; the middle of the map can be thought of as a Goldilocks region.

The quest to understand these mechanisms in biology has motivated research in the author’s group on the regulon regulators in the middle of the map. Although instinct might suggest that the middle is a place that is buffered from the difficult biological jobs and complex mechanisms found toward the corners of the map, in fact it is perhaps the most challenging place for mechanisms to meet biological demands. This is because both affinity and specificity must be very finely tuned to permit regulation.

### 4.3. So How Do They Do It?

At least in part, they cheat! Many regulon regulators solve this problem by something like cheating: their binding to DNA responds in turn to the binding of a small-molecule coeffector, often an end product of the regulated pathway. These interactions often have an allosteric nature, something that is itself a topic for an entire volume and is not discussed further here. Two examples of the roles of coeffectors are tryptophan repressor, TrpR, and arginine repressor, ArgR, which have been studied intensively in the author’s research group. Table 1 shows the results of DNA affinity and specificity measurements for the two proteins in presence and absence of their small-molecule coeffectors, each of which is the terminal product of its respective regulated pathway. In each case, the K_d_ values for operator DNA affinity have been verified using one or more of the canonical operators of the regulon, both on oligonucleotides and restriction fragments, and for non-operator DNA using random-sequence oligonucleotides and longer mixed-sequence DNAs, and all measurements reported here have been found to be consistent.

The data in Table 1 show that for both proteins, the presence of the coeffector governs not only the affinity of the protein for its DNA target, but also its specificity. This result may be surprising at first glance, at least in part because experiments to determine the four K_d_ values required to make this inference are not typically reported for other binding proteins. For ArgR, the presence of the coeffector L-arg increases operator affinity by 60-fold (K_d_ 5 nM vs. 300 nM), whereas non-operator affinity increases by 10-fold in presence of L-arg (1 µM vs. 10 µM). The fact that the two ratios are unequal already indicates that specificity is altered by the coeffector. The effect can be seen equivalently, and directly, when comparing the discrimination between operator and non-operator DNAs in presence and absence of the coeffector. The protein favors operator over non-operator by a factor of 200 in K_d_ in the presence of L-arg (5 nM vs. 1 µM), but by only a factor of 30 in the absence of L-arg (300 nM vs. 10 µM). Overall, data like these indicate that the small molecule affects the protein’s DNA binding, which is an expected conclusion. However, the data also lead to the unavoidable conclusion that DNA identity affects L-arg affinity. Although this conclusion appears at first glance to be unexpected, it is a necessary consequence of the fact that the binding equilibria for DNA and for coeffector are thermodynamically linked [2].

Because the number of such cases reported in the published literature is so few, it might be tempting to suppose that the ArgR case represents an anomaly. However, Table 1 shows precisely the same picture for TrpR, which shares no other features in common with ArgR except for being another DNA-binding regulon regulator that uses its end product as a coeffector. Here, we see that the coeffector L-trp increases operator affinity by 200-fold (K_d_ 0.5 nM vs. 100 nM) but has no effect on non-operator affinity. The fact that there is no effect of L-trp on non-operator DNA affinity shows that DNA is a coeffector equally with L-trp! In fact, it appears that K_d_ = 100 nM might be considered as something like a basal level of affinity for TrpR regardless of DNA sequence, and that only in presence of an operator DNA can the full affinity potential of the protein be expressed, and then only in the presence of L-trp. This author fervently hopes that data like those in Table 1 will be forthcoming for other proteins to enable comparative analysis.

In fact, these results also call into question our understanding of the limits of covalency in proteins [22]. Phylogenetic analysis of TrpR shows that some relatives present a tryptophan residue in exactly the position where the coeffector binds to *E. coli* TrpR [23]. Substitution of this Trp residue by Gly in the *E. coli* protein lineage created a binding site for exogenous L-trp, bringing the protein and regulon under control of the metabolic end product. A similar mutation from an Arg to Gly residue to create a binding site for L-arg has been discussed for ArgR [24]. It is a small step from these results to the inference that protein polypeptides and their monomeric amino acid constituents lie on a structural and functional continuum, and that binding sites for free amino acid ligands can evolve when mutations replace protein residues corresponding to the free amino acid. One logical but apparently untested prediction of this view is that protein surfaces are dotted with potential binding sites for free amino acids (as well as other small molecules). 

### 4.4. Molecular Mechanism and Map Position

One of the best-characterized mechanisms for tuning of affinity and specificity by a middle-map protein is the one used by *E. coli* TrpR. A partial protein-folding reaction coupled to the coeffector- and DNA-binding reactions permits modulation of both affinity and specificity. Originally it seemed that the protein had a largely disordered helix-turn-helix (HtH) domain that became all helical once bound to L-trp and DNA [25,26]. However, later evidence indicated that the HtH is in dynamic equilibrium between at least two alternative, mostly helical, conformations [27,28], only one of which permits the protein’s sidechain functional groups to occupy locations compatible with binding of DNA and the co-effector L-trp. Although ‘folding’ somewhat overstates the TrpR case, the term is used here as a shorthand for this conformational transition coupled to binding of the ligands L-trp and DNA. Other systems, including notably the MHC–peptide case, come closer to meeting what we ordinarily think of as protein folding.

### 4.5. Protein Conformational Transitions Coupled to Binding

Mechanisms related to folding are of general relevance for understanding the modulation of affinity and specificity that lies behind positions on the 2D affinity/specificity map. The coupling of a folding process to a binding process is a mechanism that can simultaneously, but independently, modulate both affinity and specificity. A key word here is ‘independently’ because it means not all such interactions will end up in the upper right corner of the map, allowing many combinations of affinity and specificity that can serve the biology of the molecules, unlike the host–guest case where accrual of affinity and specificity proceed in parallel.

When a protein is partially unfolded in its free state, and becomes folded upon binding with a partner, the effect on affinity derives from the fact that there is a thermodynamic cost for the folding process, and this cost is ‘paid’ from the binding free energy. This analysis is identical to the one described above for the linear host. However, for proteins it may seem counterintuitive because we think of native proteins as being in their lowest-energy state already. However, if in the native state a protein is partially unfolded, that means that the partially-folded state *is* the free-energy minimum. Therefore, an input of energy is required to bring the protein to the fully-folded state, and this energy comes out of (is ‘paid’ from) the ligand-binding energy.

Thus, folding coupled to binding is a mechanism that reduces affinity compared with binding to the fully-folded protein. Reducing the affinity can be especially useful for proteins that bind to DNA, because, as mentioned above, they typically contact their DNA targets over a very large surface area (e.g., ~20 base pairs for TrpR and ~40 for ArgR). Just like in the host–guest case, the release of solvent components (water, counterions, etc.) from the surface of the free protein, and even more significantly from the free DNA surface, can contribute an enormous entropic gain upon complex formation. As shown, most strikingly in the case of the *lac* repreessor, these effects can lead to extremely high affinities in vitro that would be counterproductive for the biological role of the interaction. So for many DNA-binding proteins, there may be a need to reduce affinity to the biologically effective range, and coupled folding is one way to achieve that. Besides explicit folding processes, some proteins display damping of their dynamics upon DNA binding that serves the purpose.

Specificity is also dramatically tuned by folding coupled to binding, and again the reasoning echoes the host–guest case. In the free state of a partially-folded protein, the sidechains in the binding domain are not precisely positioned in the conformations they require for close-range interactions with DNA functional groups through hydrogen-bonding or van der Waals interactions. This case resembles the linear host, where we saw that specificity was limited because the adaptations that occur upon ion binding allow enough adjustment to accommodate ions of widely different sizes. Proteins that can adapt upon binding, on the other hand, can fine-tune the close-range interactions with their DNA targets to permit the degree of discrimination among closely related sequences that is required to serve the biology of the molecules.

### 4.6. The Gestalt of Proteins

Although cooperativity is often regarded as an inscrutable emergent property, it is apparently a near-universal property of proteins [29] that is predicted to be present even in those proteins whose cooperativity has not (yet) been documented. In fact, it seems likely that proteins are designed by evolution to respond to the binding of a wide range of substances, and to do so with cooperativity. For example, Weber tabulated examples of enzymes for which cooperativity had been documented by careful quantitative study [2], even for non-natural compounds or between simple ions present in the assays used for kinetic analysis. Pyruvate kinase displays ΔΔG = −1.4 kcal/mol of cooperative free energy between binding of potassium ion and binding of manganese ion. Bovine serum albumin displays ΔΔG = +1.5 kcal/mol of cooperative free energy between binding of anilinonaphthalene sulfonate (a fluorescent probe) and binding of 3,5 dihydroxybenzoate (a product of the metabolic breakdown of lipids). 

The picture that emerges from knowledge of examples like these is that protein surfaces are bristling with sites that may accommodate a wide variety of substances, ranging from those with chemistries similar to protein constituents like amino acids to those entirely foreign to biology. It should therefore come as no surprise that many proteins respond to the binding of free amino acids, which after all are the monomeric constituents of protein polymers; as seen above this fact leads us to wonder where precisely is the boundary between covalent and non-covalent properties of proteins [22]. Evolution clearly has exploited the universality of small-molecule binding to elaborate feedback regulatory proteins that respond to the end products of the regulated operons for amino acid metabolism (e.g., tryptophan and arginine repressors, among many).

### 4.7. Cooperativity Modulates Both Affinity and Specificity

It seems also highly probable that cooperativity can modulate both affinity and specificity, forming a third dimension of the map that converts the 2D plane into a 3D cube (Figure 1B). Experimental evidence supports this view in the case of the TrpR protein–DNA interaction. On synthetic operators that bind a pair of TrpR dimers in a tandem arrangement, the affinity and the apparent cooperative free energy between dimers vary with DNA sequence and sequence context in a manner that suggests there is compensation between affinity and cooperativity to maintain DNA occupancy at a level appropriate for regulation [30]. A compensation mechanism of this kind allows the range of DNA sequences found among the various operators of the regulon to be bound with similar effectiveness. The ability to accommodate sequence variation may be necessary in regulatory regions like this one that serve dual purposes, such as overlapping operators and promoters.

### 4.8. About Nonspecific DNA

For a nonspecific DNA to be meaningfully used for determining ΔΔG its affinity for the binding protein must be sequence-independent—in fact, that can be thought of as a definition of nonspecific DNA binding. In practice, there are several ways to ensure this criterion is met, but not all of them are equal. For example, readily available homopolymers like polydA-polydT or polydG-polydC might be thought to provide ideal nonspecific targets. There are at least two problems with using such targets. First, any possible sequence-dependent effects that can affect binding affinity may be amplified in a homopolymer. Second, the molar concentration of a polymer is typically given in units of moles of base pairs, not moles of molecules, because the extent of polymerization may be unknown or variable or both. This concern applies as well to mixed-sequence DNA polymers like salmon sperm DNA. With different units of concentration it is meaningless to compare the ΔG value for binding of a polymer with the ΔG value for a discrete DNA molecule like an oligonucleotide that represents an operator target.

A better way to evaluate specificity for DNA-binding proteins is to use a discrete oligonucleotide of length similar to that of the operator target, but what sequence to choose? If a determined value of ΔΔG truly represents the specificity of a given protein for its operator, then ΔΔG should be the same regardless of the sequence of nonspecific DNA used. Therefore, the best way to achieve this condition is to measure the affinities of several different nonspecific DNAs. The base composition is unlikely to be important, but it is essential that no partial sequences, even as short as dinucleotide pairs [31], are shared with the operator target. It is also essential that the specific and nonspecific DNAs be evaluated using the same experimental methods and the same DNA concentrations.

Another alternative worth mentioning is even more readily deployed for a useful rough estimate. Figure 3 depicts a strategy for restriction cleavage and labeling that generates two 5′-end-labeled fragments of similar but distinguishable lengths, one with and the other without the target site for the binding protein. The fragments need not be purified from the restriction digest before titrating with the binding protein and resolving the bound and free DNAs by electrophoretic mobility shift analysis [21]. It is important to ensure that the non-operator fragment contains no sequences related to the operator fragment. A variant of this approach that may better approach sequence randomization and that avoids labeling is to prepare a restriction digest containing numerous fragments of distinguishable lengths, titrate the unresolved fragment mixture with the binding protein, and resolve the bound and free species by electrophoretic mobility shift with detection by staining of the DNA [32]. Many DNA-binding proteins display length-dependent affinity that is sequence-independent, a feature that is readily detected in this experiment if present. It is also essential to remember that whenever detection methods require more than trace amounts of DNA, the DNA concentration can affect the apparent affinity [33,34].

*Bridging the in vitro—in vivo gap.* Although as a rule this gap is insurmountable quantitatively for reasons given above, there is one special case where it can be addressed at least in relative quantitative terms. Somewhat surprisingly, actively growing cells readily take up dimethyl sulfate, a compound that modifies intracellular DNA and can be used for footprinting of binding proteins in vivo [35]. The chemical reaction is methylation of guanine residues on the major groove side of the base, and of adenine residues on the minor groove side of the base e.g., [36]. A bound protein occludes the bases in its binding site and can thus be detected as a site of diminished methylation relative to the surrounding unprotected sites. 

To use in vivo footprinting semi-quantitatively requires some knowledge of, and/or control over, the intracellular concentration of the binding protein. These requirements are seldom met. However, it is possible to vary the protein concentration to some extent if it can be expressed at differing levels by using either a series of plasmids of varying copy number or a controllable expression system. Although this approach will not yield exact (or necessarily knowable) protein concentrations, it will permit estimation of relative affinities (occupancies) of different operators simultaneously at each protein concentration. The author’s group used this approach to evaluate the number and relative affinities of tryptophan repressor dimers occupying the chromosomal operator sites in each of the operons of its regulon [30]. The results agreed in detail with those determined in vitro, and in addition identified the probable presence of bound RNA polymerase adjacent to the bound repressor in one site. These approaches should be more widely used in efforts to bridge the gap between in vitro and in vivo analysis of protein–DNA interactions.

## Figures and Tables

**Figure 1 ijms-23-00562-f001:**
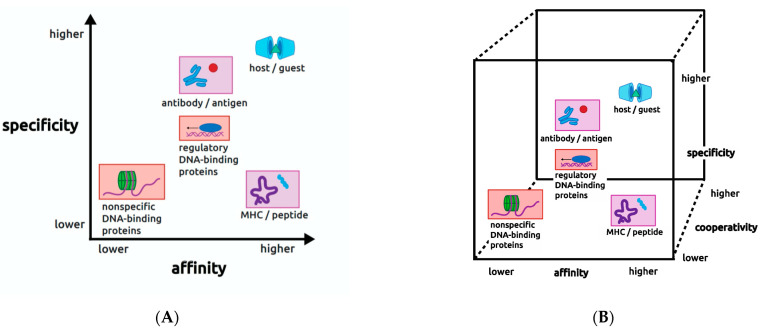
(**A**) The affinity/specificity plane. (**B**) Cooperativity modulates both affinity and specificity. Cooperativity forms a third axis of the affinity/specificity plane, converting it to a cube. Figure prepared by Ethan Sample.

**Figure 2 ijms-23-00562-f002:**
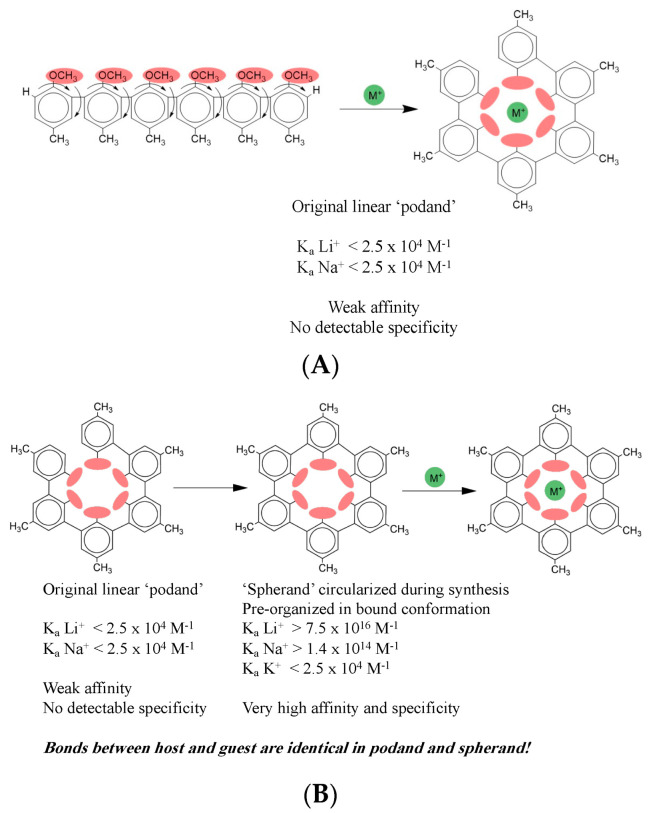
(**A**) Early host–guest example using linear podand host. (**B**) Pre-organized host–guest example. Figure prepared by Ethan Sample.

**Figure 3 ijms-23-00562-f003:**
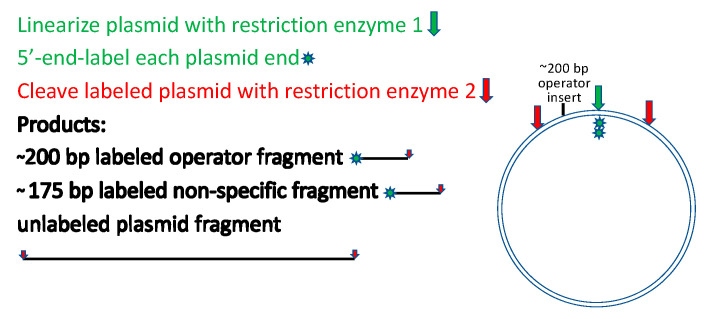
Labeling strategy for simultaneous analysis of specific and nonspecific binding.

**Table 1 ijms-23-00562-t001:** Influence of small-molecule coeffectors on affinity and specificity.

		Dissociation Constant, K_d_	
		Operator DNA	Non-Operator DNA
Arginine repressor, ArgR [20]			
+L-arginine		5 nM	1 µM
-L-arginine		300 nM	10 µM
Tryptophan repressor, TrpR [21]			
+L-tryptophan		0.5 nM	100 nM
-L-tryptophan		100 nM	100 nM

## Data Availability

Not applicable.

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
