# Peer review of "Affinity, Specificity, and Cooperativity of DNA Binding by Bacterial Gene Regulatory Proteins"

_ijms, 2022, doi:10.3390/ijms23010562_

Round 1

Reviewer 1 Report

The manuscript, originally, introduces the host-guest chemistry, getting through the human immune system and the thermodynamics of molecular interactions chemistry to disentangle affinity and specificity in biology. It would be worth trying to summarize and relate these concepts to the operon regulatory logic in a discussion section.

In addition, I’m wondering whether does exist in the opinion of the author a correlation between DNA looping and “pre-organized” host-guest chemistry mentioned.

Regarding the thermodynamics implications of Protein-DNA interactions a comment to the manuscript “Two distinct DNA sequences recognized by transcription factors represent enthalpy and entropy optima”. (Elife. 2018 Apr 11;7:e32963. doi: 10.7554/eLife.32963. by Morgunova E, Yin Y, Das PK, Jolma A, Zhu F, Popov A, Xu Y, Nilsson L, Taipale J) may help putting the manuscript in the context of current literature. Along this line, considering the breadth of the arguments recalled by the author, I’m also wondering whether it would be possible to correlate these concepts with those of current scientific debate on transcriptional noise and phase separation. Finally, concluding the manuscript with some open outstanding questions in my opinion would underscore and strengthen the relevance of the principles recalled.

Author Response

  1. It would be worth trying to summarize and relate these concepts to the operon regulatory logic in a discussion section. To the limited extent such relating is possible it is included in the section, The special case of regulatory protein-DNA interactions.
  2. In addition, I’m wondering whether does exist in the opinion of the author a correlation between DNA looping and “pre-organized” host-guest chemistry mentioned. This is an interesting extrapolation but i don't think that DNA looping can be considered as a literal, macro-scale representation analogous to the closed host structure.
  3. Regarding the thermodynamics implications of Protein-DNA interactions a comment to the manuscript “Two distinct DNA sequences recognized by transcription factors represent enthalpy and entropy optima”. (Elife. 2018 Apr 11;7:e32963. doi: 10.7554/eLife.32963. by Morgunova E, Yin Y, Das PK, Jolma A, Zhu F, Popov A, Xu Y, Nilsson L, Taipale J) may help putting the manuscript in the context of current literature. The article cited is one of innumerable articles in which thermodynamics of protein-DNA interactions have been discussed. Nearly all such articles suffer from a limitation addressed in the ms text, namely that any attempt at attribution of enthalpic or entropic contributions neglects the many cryptic sources that often dominate the overall thermodynamics.
  4. Finally, concluding the manuscript with some open outstanding questions in my opinion would underscore and strengthen the relevance of the principles recalled. Although the ms does not conclude with such questions, they are articulated in the abstract and introduction, which constitute a call to apply the outlined principles to other protein-DNA systems. The final section of the ms on bridging the in vivo and in vitro gap, illustrates to readers how the principles can be applied to any other system.

Reviewer 2 Report

This approach evaluate the number and relative affinities of tryptophan inhibitory dimers occupying chromosomal manipulator sites in each operon of that regulon. Semiquantitative use of the in vivo footprint requires some knowledge and/or control over the intracellular concentration of the binding protein. The results were in close agreement with the results determined in vitro, and also confirmed the possible presence of bound RNA polymerase adjacent to the bound repressor at one site. This review paper is well written. However, the scope of the related papers seems to be somewhat lacking. Please update the reference section to the latest version for reference.

Author Response

  1. This approach evaluate the number and relative affinities of tryptophan inhibitory dimers occupying chromosomal manipulator sites in each operon of that regulon. ... However, the scope of the related papers seems to be somewhat lacking. Please update the reference section to the latest version for reference. The first sentence suggests that the reported results were taken to be current research results, rather than a published example used to illustrate underlying principles. As such, the scope of papers referenced include those that are appropriate and necessary for the purpose of the manuscript.